# Trends in Beef Intake in the United States: Analysis of the National Health and Nutrition Examination Survey, 2001–2018

**DOI:** 10.3390/nu15112475

**Published:** 2023-05-26

**Authors:** Clara S. Lau, Victor L. Fulgoni, Mary E. Van Elswyk, Shalene H. McNeill

**Affiliations:** 1National Cattlemen’s Beef Association, a contractor to the Beef Checkoff, 9110 East Nichols Ave., Suite 300, Centennial, CO 80112, USA; smcneill@beef.org; 2Nutrition Impact, LLC, Battle Creek, MI 49014, USA; vic3rd@aol.com; 3Van Elswyk Consulting, Inc., Clark, CO 80428, USA; mveconsulting@q.com

**Keywords:** red meat, beef, lean fresh beef, ground beef, processed beef, dietary intake, National Health and Nutrition Examination Survey (NHANES), United States, usual intake, meat products

## Abstract

Evidence-based dietary advice regarding meats (including beef), requires accurate assessment of beef and other red meat intakes across life stages. Beef intake is subject to misclassification due to the use of broad categories such as “red and processed meat”. In the current study, intake trends for total beef (i.e., any beef type) and specific beef types (fresh lean, ground, processed) among Americans participating in the National Health and Nutrition Examination Survey (NHANES) 2001–2018 (*n* = 74,461) were characterized and usual intake was assessed using NHANES 2011–2018 (*n* = 30,679). The usual intake amounts of beef were compared to those of relevant protein food subgroups modeled in the Healthy U.S.-Style Dietary Pattern (HDP) reported in the 2020–2025 Dietary Guidelines for Americans (DGA). Total per capita beef consumption declined an average of 12 g (*p* < 0.0001) for ages 2–18 years and 5.7 g (*p* = 0.0004) for ages 19–59 years per 2-yr NHANES cycle, over the 18-year timeframe, while remaining unchanged for Americans aged 60+ years. On a per capita basis, Americans aged 2 years and older consumed 42.2 g (1.5 ounces) of total beef per day. Fresh lean beef per capita consumption was 33.4 g (1.2 ounces) per day. Per capita intake was similar across all age groups and below the daily HDP modeled amount of 3.7 ounce equivalents for the “Meats, Poultry, Eggs” (MPE) subgroup, while approximately 75% of beef consumers’ intakes of total beef was within HDP modeling. Evidence from intake trends suggests beef is not overconsumed by the majority of Americans but rather within the amounts for MPE and red meat modeled in the HDP of the DGA at the 2000-calorie level.

## 1. Introduction

Beef is a commonly consumed food and an inherent source of many essential nutrients including high-quality protein, vitamins B6 and B12, zinc, and readily bioavailable heme iron [1,2]. Beef is a red meat, but the commonly used phrases “red meat” and “red and processed meat” encompasses more than just beef, typically referring to the combination of beef, pork, lamb, and game meat, both fresh and processed [3]. The use of broad terminology such as “red and processed meat” increases the risk of misclassification of beef and other red meat intake data [4,5]. Nonetheless, recommendations to eat less “red and processed meat” persist in current dietary guidance in the United States (U.S.) [6,7] based, at least in part, on observational evidence reporting weak associations between “red and processed meat” consumption and risk of chronic disease [8,9]. Interpretation of advice to eat less “red and processed meat” is further complicated by advice that encourages “lean meat” [6,7], as lean beef, red meat, processed meat, and lean meat are not mutually exclusive categories [5]. The Dietary Guidelines for Americans, 2020–2025 (DGA) indicate that protein foods, including lean meats, are core elements in a healthy diet [7]. Although the DGA do not provide beef-specific intake recommendations, the Healthy U.S.-Style Dietary Pattern (HDP), for a 2000-calorie diet, models 3.7 ounces (oz) or 104.9 g of lean meat, poultry and/or eggs per day as part of a healthy dietary pattern [7].

Without a comprehensive and accurate knowledge of intake of individual meat types across all life stages, it is difficult to evaluate the relationships between red and processed meat intake and disease risk to support the development of evidence-based dietary advice [4]. Previous analyses of National Health and Nutrition Examination Survey (NHANES) 1999–2016 reported the consumption of processed meat (including processed red meat and processed poultry meat) remained unchanged, while fresh red meat and fresh beef consumption decreased among adults in the U.S. [10]. During this same time period, fresh poultry, predominantly chicken, significantly increased among U.S. adults [10]. The current study characterizes intake of total beef (i.e., any beef type) and individual beef types (fresh lean, ground, and processed) across life stages using NHANES 2001–2018 data, with the specific objectives to: (1.) Describe intake trends for total beef and individual beef types for the general population across life stages; (2.) Report intake distribution of beef and beef types on both a per capita and per consumer basis; and (3.) Compare per capita and per consumer beef intake to the modeled “Meats, Poultry, Eggs” (MPE) subgroup of the Protein Foods group in the HDP presented in the 2020–2025 DGA.

## 2. Materials and Methods

### 2.1. Dietary Intake Assessment

The NHANES is a program of the National Center for Health Statistics which is part of the Centers for Disease Control and Prevention and has the responsibility for producing vital and health statistics for adults and children living across the U.S. The nutrition examination aspect of the survey, What We Eat in America (WWEIA), relies on the U.S. Department of Agriculture (USDA)’s Automated Multiple-Pass Method, a computer-assisted multiple-pass format interview system with standardized probes, developed by the USDA to estimate current dietary intake and to minimize misclassification of dietary intake data of foods [11]. A detailed description of the subject recruitment, survey design, and data collection procedures for NHANES are available online at https://www.cdc.gov/nchs/nhanes/about_nhanes.htm (accessed on 23 April 2021).

NHANES has stringent consent protocols and procedures to ensure confidentiality and protection from identification. The present study was a secondary data analysis of publicly available de-identified data and therefore does not meet the criteria for human subjects research and therefore was exempt from additional approvals by institutional review boards. All participants provided a signed written informed consent. All data obtained from this study are publicly available at: http://www.cdc.gov/nchs/nhanes/ (accessed on 23 April 2021).

### 2.2. Analytical Sample—Beef Intake Trends

For analyses examining NHANES cycle-to-cycle trends in beef intake, 24-h dietary recall data from the in-person interview (Day 1) from 74,461 subjects aged 2+ years (after exclusions for unreliable or incomplete data *n* = 10,163, and subjects with no Day 1 dietary intake data *n* = 5) participating in NHANES 2001–2018 were used. Data were analyzed separately for the following age groups: 2–18 years (children and adolescents), 19–59 years (adults), and 60+ years (older adults), as outlined in the DGA.

### 2.3. Analytical Sample—Usual Intake of Beef, per Capita and per Consumer

For analyses looking at current beef intake distribution, we used data from NHANES 2011–2018, which provides an updated assessment of previous work [1,12]. The total sample included 30,679 subjects after exclusions. Usual intake was determined using the National Cancer Institute’s (NCI) methodology [13] incorporating both days of dietary recall. The two-part model (proportion and amount) was used to determine usual intakes for both per capita (combines consumers and non-consumers of beef) and consumer-only population groups (defined as those with beef consumption on Day 1). Data representing the combination of consumers and non-consumers of beef are referred to as “per capita” in the results and discussed using “average”-related terminology. Data representing consumer-only populations are referred to as “beef consumers” or “consumer-only”. Data were analyzed separately for age groups: 2+ years (general population), 2–18 years (children and adolescents), 19–59 years (adults), and 60+ years (older adults), as outlined in the DGA.

### 2.4. Calculation of Beef-Related Variables

The Food and Nutrient Database for Dietary Studies (FNDDS) is used to code and analyze dietary intakes collected in the WWEIA portion of the NHANES survey (https://www.ars.usda.gov/northeast-area/beltsville-md-bhnrc/beltsville-human-nutrition-research-center/food-surveys-research-group/docs/fndds/; accessed on 23 April 2021). FNDDS food code ingredient data were used to determine the ingredient profile of all food codes consumed. The USDA Food Patterns Equivalent Database (FPED) and the Food Pattern Ingredient Database (FPID) are linked to the food codes and ingredient codes were used to quantify beef content (https://www.ars.usda.gov/northeast-area/beltsville-md-bhnrc/beltsville-human-nutrition-research-center/food-surveys-research-group/docs/fped-overview/; accessed on 23 April 2021). Four types of beef (total beef, ground beef, fresh lean beef, processed beef) were analyzed (Table 1) and calculated separately [3]. A similar approach to reconcile meat sources in multiple meat-ingredient foods was used by O’Connor and colleagues [14]. For beef-related variables determined to be entirely beef, the percentage of beef was assumed to be 100%. For foods/ingredients that contain beef where a second type of meat was indicated by the description, e.g., “bologna, beef and pork”, then the percentage used for beef was 50%. Similarly, for foods/ingredients containing beef, where the description suggests that at least two other types of meat were included the percentage contribution by beef was assumed to be 33%. Beef types are not mutually exclusive, so the types will not equal the total, e.g., fresh lean beef includes a portion that is ground, but not all ground beef is lean, and some consumers only consume certain beef types. Furthermore, lean beef is calculated using a USDA-established threshold of 2.63 g solid fat or less per ounce equivalent to qualify as lean beef, with solid fat in excess of 2.63 g allocated to the solid fat component of FPED (Table 1) [3]. Consequently, subject numbers for the total beef category will reflect consumers of both lean and higher fat beef.

### 2.5. Comparison of Usual Intake of Beef to “Meats, Poultry, Eggs” Modeled in HDP

In an effort to assess beef intake against dietary advice, data in the current study were compared to the MPE subgroup modeled in the HDP as reported in the DGA [7]. The DGA notes that “Meats include beef, goat, lamb, pork, and game meat (e.g., bison, moose, elk, deer)” and that meats and poultry should be lean or low-fat [7]. Food groups used in the HDP modeling are composed of the most nutrient-dense forms of foods (i.e., prepared with the lowest amounts of sodium, saturated fat and added sugars) [15]. Thus, it is understood that meats in the HDP model represent lean red meat. Appendix A describes the steps used to derive the daily amounts of MPE, collectively, and the amount of lean red meat, individually, as modeled in the HDP and used as comparison points in the current analysis. USDA utilizes food group and subgroup item clusters to complete dietary pattern modeling [15]. Item clusters are groupings of similar foods, in nutrient-dense form, used to calculate the nutrient profile of a food group or subgroup [15]. Proportional intakes of an item cluster based on a composite population-weighted average intake of the general U.S. population and/or varying life stage populations are the benchmarks for USDA dietary pattern models [15,16]. Using this approach, 3.7 ounce equivalents are modeled for the MPE of the HPD at the 2000-calorie level (Appendix A). To determine the amount of lean red meat modeled in the 3.7 ounce equivalents of MPE of the HDP, the USDA protein food item cluster was disaggregated on a percent contribution basis into the individual representative foods (Appendix A). The amount of lean meat in the protein food item cluster was then totaled and used as the denominator for red meat in the red meat food subgroup, resulting in 1.8 ounce equivalents of lean red meat per day or 12.5 ounce equivalents per week. For the purposes of comparison to the HDP, it is noted that ounce equivalents and ounces are synonymous for lean meat [7]. Further, as noted in Table 1, fresh lean beef data excludes fat contribution in excess of that considered lean (9.28 g fat per 100 g) and is therefore synonymous with lean meat ounce equivalents, as modeled in the HDP. Total beef data will include solid fat in excess of 9.28%.

In the current analysis, we compared both per capita and consumer-only usual intake levels of beef, to that allocated in the modeled amount of MPE in the 2000-calorie level of the HDP, as a way to establish a reference point. Modeling of the HDP involves establishing food groups and food group amounts, in part, based on consumption-weighted nutrient dense food averages over time [15]. Thus, on any given day, consumers might choose only one (e.g., beef) or a few food sources of protein in the MPE group, but over time consumers might end up consuming all the MPE components (such as after a week or so). This also allows allocation of an entire day’s MPE modeled allowance to beef, as practically speaking, not everyone includes all components of the MPE subgroup in their daily diet. Additionally, the HDP does not distinguish beef from pork and other red meat, and at the 2000-calorie level of the HDP, 1.8 of the 3.7 oz of MPE is allocated to lean red meat (i.e., both fresh and processed).

### 2.6. Statistical Analysis

Analyses were performed using SAS 9.4 and data adjusted for the complex sampling (clustered sample) design of NHANES, using appropriate survey weights, strata, and primary sampling units. Least-square means and the standard errors were calculated using regression analyses adjusted for key covariates (age, gender and ethnicity). A *p*-value of <0.05 was deemed significant. To assess intake over time, NHANES cycles were numbered 1–9 for 2001–2002 through 2017–2018, respectively; the regression coefficient generated by these analyses generates a change per NHANES cycle (e.g., every two years). Distribution of usual intakes of beef was also determined using the NCI programs [17,18] on a per capita basis and on a consumer-only basis. Using the distribution of usual intake, we assessed the percentage of the population exceeding certain levels of beef intake. Additionally, we assessed the source-of-food variable to ascertain where subjects were getting ground beef.

## 3. Results

### 3.1. Trends in Intake of Beef by Age Group, NHANES 2001–2018

Children and adolescents between 2 and 18 years of age consumed significantly less total beef in 2018 as compared to 2001 (β = −1.66 g/cycle, *p* < 0.0001), averaging 12 g (0.4 oz) less per day from 2001–2002 (41.9 ± 1.5 g/day (1.5 ± 0.05 oz/day)) compared to 2017–2018 (30.0 ± 2.5 g/day (1.1 ± 0.1 oz/day), Figure 1). Declines in the consumption of each beef type (i.e., fresh lean, processed, ground) contributes to the observed decrease in total beef consumption (β = −1.22 g/cycle, *p* < 0.0001; β = −0.80 g/cycle, *p* < 0.0001; β = −0.33 g/cycle, *p* = 0.0005, respectively).

Compared to 2001, total beef intake in adults aged 19–59 years was an average of 5.7 g (0.2 oz) less per day in 2018 (β = −1.14 g/cycle, *p* = 0.0004). Of beef types, intakes of fresh lean and processed beef contributed to the significant decline (β = −0.68 g/cycle, *p* = 0.0206 and β = −0.39 g/cycle, *p* < 0.0001, respectively) in total beef intake among adults while ground beef intake remained relatively consistent (β = −0.38 g/cycle, *p* = 0.08).

Older adults aged 60 years and older, on average, maintained their intake of beef over the nine NHANES cycles (β = −0.05 g/cycle, *p* = 0.8866). Currently, adults aged 60+ years are reported to consume 41.8 ± 3.5 g/day (1.47 ± 0.1 oz/day) of beef compared to 38.3 ± 0.9 g/day (1.35 ± 0.03 oz/day) reported in 2001–2002. Intake of all beef types remained consistent over time.

### 3.2. Per Capita Usual Intake of Beef, NHANES 2011–2018

On average, Americans 2 years and older (*n* = 30,679) consumed 42.2 ± 0.9 g (1.5 ± 0.03 oz) of total beef each day. Fresh lean beef intake was reported as 33.4 ± 0.8 g (1.2 ± 0.03 oz) per day (Appendix A).

Children and adolescents between 2 and 18 years of age (*n* = 10,913) consumed, on average, 31.9 ± 0.9 g (1.1 ± 0.03 oz) of total beef per day. This age group consumed 22.9 ± 0.8 g (0.8 ± 0.03 oz) of fresh lean beef per day (Figure 2, Appendix A). More specifically, mean usual intake of total beef of males aged 2–18 y was 36.6 ± 1.3 g (1.3 ± 0.04 oz) and of females aged 2–18 y was 27.0 ± 1.0 g (1.0 ± 0.03 oz) per day. Fresh lean beef consumption by these age and gender groups was 26.2 ± 1.1 g (0.9 ± 0.04 oz) and 19.5 ± 0.8 g (0.7 ± 0.03 oz) per day, respectively.

Adults aged 19–59 years (*n* = 13,203) consumed, on average, 47.1 ± 1.1 g (1.7 ± 0.04 oz) of total beef per day. Adults in this age group consumed 38.2 ± 1.0 g (1.4 ± 0.04 oz) of fresh lean beef per day (Figure 2, Appendix A). More specifically, mean usual intake of total beef of males aged 19–59 years was 60.8 ± 1.7 g (2.1 ± 0.1 oz) and of females aged 19–59 years was 33.1 ± 1.1 g (1.2 ± 0.04 oz) per day. Fresh lean beef consumption by these age and gender groups was 49.0 ± 1.5 g (1.7 ± 0.1 oz) and 27.1 ± 1.0 g (1.0 ± 0.03 oz) per day, respectively.

Older adults aged 60+ years (*n* = 6563) consumed, on average, 40.7 ± 1.2 g (1.4 ± 0.04 oz) of total beef per day. Fresh lean beef consumption by older adults was 32.0 ± 1.2 g (1.1 ± 0.04 oz, Figure 2, Appendix A). More specifically, mean usual intake of total beef of males aged 60+ years was 51.5 ± 2.1 g (1.8 ± 0.1 oz) per day and of females aged 60+ years it was 31.6 ± 1.3 g (1.1± 0.04 oz) per day. Fresh lean beef consumption by males and females 60+ years was reported to be 40.3 ± 2.1 g (1.4 ± 0.1 oz) and 25.1 ± 1.2 g (0.9 ± 0.04 oz) per day, respectively.

Of the beef types reported, processed beef was the least consumed, with mean usual intakes of 7.5 ± 0.4 g (0.3 ± 0.01 oz) per day by 2–18 years, 6.4 ± 0.3 g (0.2 ± 0.01 oz) per day by 19–59 years, and 6.6 ± 0.5 g (0.2 ± 0.02 oz) per day by 60+ years groups (Figure 2, Appendix A). Between 2001 and 2018, processed beef intake significantly decreased in the total population, specifically in the 2–18 years and 19–59 years age groups, while remaining constant in the 60+ years age group (Figure 1). Mean usual intakes of ground beef were 16.0 ± 0.6 g (0.6 ± 0.02 oz) per day by 2–18 years, 22.7 ± 0.8 g (0.8 ± 0.03 oz) per day by 19–59 years, and 17.1 ± 0.7 g (0.6 ± 0.03 oz) per day by 60+ years groups (Figure 2, Appendix A). On average, ground beef consumed per capita was from sources other than fast food (Appendix A).

### 3.3. Comparison of Per Capita Usual Intake of Beef to HDP Modeling

Intake distribution of per capita usual intake of total beef and fresh lean beef from NHANES 2011–2018 is reported in Figure 3. All age groups are eating within the 3.7 ounce MPE based on the 2000-calorie level, as modeled in the HDP (Figure 3). On a per capita basis, 82% of the 2–18 years, 62% of the 19–59 years, and 77% of the 60+ years age groups reported intakes of total beef at or below the 1.8 oz of red meat level, as modeled in the HDP at the 2000-calorie level. Additionally, 95% of the 2–18 years, 78% of the 19–59 years, and 92% of the 60+ years age groups are consuming fresh lean beef at levels at or below the 1.8 oz of lean red meat modeled in the HDP at the 2000-calorie level.

### 3.4. Beef Consumer Usual Intake of Beef, NHANES 2011–2018

On the day of recall, approximately half (50.4%) of respondents consumed beef and will be referred to as “beef consumers”. Similarly, approximately half of the 2–18 years and 19–59 years subpopulations are beef consumers (52.3% and 51.2%, respectively), while a slightly smaller proportion of the 60+ years subpopulation are beef consumers (45.3%). Data below indicate a higher number of total beef consumers than individual beef types (e.g., not all beef consumers consume all beef types, but all beef consumers consume beef) and because, as noted in the methodology section, total beef intake values are reflective of only the lean portion of beef plus 2.63 g of solid fat per ounce equivalent; therefore, the amounts of total beef and lean beef will be similar. In other words, while the sample sizes reflect the number of consumers of total beef and each beef type, the amount of solid fat in excess of 2.63 g per ounce equivalent found in higher fat beef cuts is allocated to the solid fat component of FPED. The usual intake of beef in beef consumers aged 2 years and older (*n* = 15,449), was 83.2 ± 0.9 g (2.9 ± 0.03 oz) of total beef each day (Appendix A). Usual intake of fresh lean beef of beef consumers (*n* = 11,876) 2 years and older was 83.4 ± 1.0 g (2.9 ± 0.04 oz) per day.

For beef consumers in the 2–18 years age group, the usual intake of total beef was 62.0 ± 1.7 g (2.2 ± 0.1 oz) per day; and the usual intake of fresh lean beef was 64.2 ± 1.8 g (2.3 ± 0.1 oz, Figure 4, Appendix A) per day. More specifically, usual intake of total beef of males aged 2–18 years was 68.4 ± 2.3 g (2.4 ± 0.1 oz) per day and of females aged 2–18 years it was 54.5 ± 1.6 g (1.9 ± 0.1 oz) per day. Usual intake of fresh lean beef by these age and gender groups was 71.2 ± 2.8 g (2.5 ± 0.1 oz) and 55.9 ± 1.9 g (2.0 ± 0.1 oz) per day, respectively.

For adult beef consumers aged 19–59 years, the usual intake of total beef was 91.8 ± 1.2 g (3.2 ± 0.04 oz) per day. Fresh lean beef consumption by adults under 60 years was reported to be 91.2 ± 1.4 g (3.2 ± 0.1 oz, Figure 4, Appendix A) per day. More specifically, mean usual intake of total beef of adult males was 109.0 ± 1.9 g (2.4 ± 0.1 oz) per day and of adult females it was 70.2 ± 1.5 g (2.5 ± 0.1 oz) per day. Usual intakes of fresh lean beef by these age and gender groups was 107.4 ± 2.0 g (3.8 ± 0.1 oz) and 70.6 ± 1.5 g (2.5 ± 0.1 oz) per day, respectively.

For older adult beef consumers, aged 60+ years, the usual intake of total beef was 84.1 ± 1.6 g (3.0 ± 0.1 oz) per day. Usual intake of fresh lean beef intake by older adults was 80.9 ± 1.9 g (2.9 ± 0.1 oz, Figure 4, Appendix A) per day. More specifically, usual intake of total beef of older adult males was 98.3 ± 3.1 g (3.5 ± 0.1 oz) per day and of older adult females was 70.4 ± 1.9 g (2.5 ± 0.1 oz) per day. Usual fresh lean beef intake by these age and gender groups was 95.2 ± 3.2 g (3.4 ± 0.1 oz) per day and 67.3 ± 2.1 g (2.4 ± 0.1 oz) per day, respectively.

Of the beef types reported in beef consumers, processed beef was the least consumed, with mean usual intakes of 30.8 ± 1.4 g (1.1 ± 0.1 oz) per day in 2–18 years, 38.8 ± 1.6 g (1.4 ± 0.1 oz) per day in 19–59 years, and 48.1 ± 2.9 g (1.7 ± 0.1 oz) per day in 60+ years groups (Figure 4, Appendix A). Ground beef consumers make up approximately a quarter of each age group subpopulation, with mean usual intakes of 57.3 ± 1.5 g (2.0 ± 0.1 oz) per day in 2–18 years, 76.6 ± 2.4 g (2.7 ± 0.1 oz) per day in 19–59 years, and 71.5 ± 3.0 g (2.5 ± 0.1 oz) per day in 60+ years groups (Figure 4, Appendix A). Among ground beef consumers, 28–39% report consuming ground beef from fast food sources, with mean intakes of 60.5 ± 2.5 g (2.1 ± 0.1 oz) per day in 2–18 years, 74.4 ± 2.0 g (2.6 ± 0.1 oz) per day in 19–59 years, and 65.6 ± 4.2 g (2.3 ± 0.2 oz) per day in 60+ years groups (Appendix A).

### 3.5. Comparison of Beef Consumer Usual Intakes to HDP Modeling

The distribution of beef consumers’ usual intakes of total beef and fresh lean beef are reported in Figure 5, separated by age group (2–18 years, 19–59 years, and 60+ years). If a beef consumer (age 2 years and older) chooses to consume their entire MPE amount in the HDP at the 2000-calorie level, as beef, the mean usual intakes of total beef and fresh lean beef are within the collectively modeled level of 3.7 oz. Most beef consumers included at least 1.8 oz of total beef per day with 38% of the 2–18 years, 5% of the 19–59 years, and 6% of the 60+ years groups reporting intakes at or below the 1.8 oz of red meat level, as modeled in the HDP at the 2000-calorie level. On average, daily intake levels of beef consumers are within the MPE amount, with 7% of the 2–18 years, 31% of the 19–59 years, and 19% of the 60+ years groups reporting intakes at or above the 3.7 oz MPE, as modeled in HDP at the 2000-calorie level. Less than 2% of adults (19–59 years) consume total beef above the 5.5 ounce equivalents for the protein foods group (which includes MPE, seafood, nuts, seeds, and soy products), as modeled in the HDP.

## 4. Discussion

Americans’ consumption of beef has significantly declined over the most recent nine cycles (18-year timeframe) of NHANES data. Our findings of declining trends of beef consumption are consistent with Kim et al. [19] and Zeng et al. [10], who reported significant declines in Americans’ consumption of total beef in adolescents (age 12–19 years) and of unprocessed beef in adults (20 years and older) based on older NHANES survey data.

The current data analyses indicate that, on average, the majority of Americans consume beef within the amounts for MPE and red meat modeled in the HDP at the 2000-calorie level. Approximately 75% of beef consumers are reported to consume total beef within the modeled amount for MPE in the HDP at the 2000-calorie level, challenging the perception that red meat is “over-consumed” [20,21,22]. More specifically, beef consumers favor fresh lean beef, typically consuming above the modeled amount of 1.8 oz of red meat but below the modeled amount of 3.7 oz for MPE, in their dietary patterns.

Commonly reported associations between consumption of red and processed meat with increased risk of chronic diseases such as cardiovascular/cardiometabolic disease have raised concerns that beef may be over-consumed [23,24]. However, in recent systematic reviews, authors conclude that red and processed meat consumption is likely not causally related to the development of cardiovascular disease [25] as the magnitude of association is very small and the evidence is of low certainty [26]. Similarly, in another study, researchers noted that the evidence that the association between unprocessed red meat intake with increased risk of disease incidence and mortality was weak and insufficient to make strong or conclusive dietary recommendations [27].

Concerns regarding “over-consumption” of beef may also relate to broader concerns regarding consumption of protein in excess of the minimum level established by the Recommended Dietary Allowance (RDA) [28,29]; however, the consumption of total protein, as a percentage of total energy intake, ranges from 14% to 16% across all age groups, well within the Acceptable Macronutrient Distribution Ranges (AMDR) of 10–35% of total energy intake. Less than 1% of Americans consume protein above the AMDR [30]. The HDP is modeled with protein contributing 18% to the total calories, while the DGA “are designed to meet the RDA and Adequate Intakes for essential nutrients, as well as AMDR, all set by the National Academies,” where protein can contribute up to 35% of total calories [7]. Thus, though the modeled amounts in the HDP serve as comparison points for this current analysis, there is flexibility for the individual to consume varying amounts of carbohydrate, protein, and fat, within a healthy eating pattern. Adolescent females (age 14–18 years) were among the most common found to have protein levels below the Estimated Average Requirement (EAR), and adult females aged 19+ years are more likely to fall below the EAR and the RDA for protein compared to adult males of the same age range [30].

The DGA encourages individuals to have the flexibility to choose a healthy dietary pattern within calorie limits that fits their personal preferences and cultural traditions [7]. With the HPD modeled at 18% of calories as protein, there is flexibility to include a higher proportion of total calories as protein (including higher protein as beef), while still within the AMDR, which previous research has demonstrated can support improved health benefits including but not limited to weight management, physical function, and heart health [31,32,33,34]. In a recent analysis, researchers demonstrated the feasibility of modifying the amount of protein within the HDP by up to 25% of total calories while meeting all nutrient needs, using the framework implemented by the USDA in developing the HDP [16]. This supports the notion that individuals can have the flexibility to enjoy a variety of protein foods, including beef, and including intake of amounts greater than those modeled in the HDP.

Beef is a commonly consumed whole-food source of high-quality protein that supports healthy aging [35,36]. In the present study, beef consumption of older adults aged 60+ did not decrease over the past 18 years. However, the reported usual intake in older adults is lower than that in the adult group, which suggests beef intake decreases as a person ages, which may lead to a decrease in their intake of high-quality protein and other essential nutrients [30,37]. There is consensus between international expert groups that as people age, there should be an increase in protein intake, as physiological needs change with aging [38,39]. For example, previous research by An and colleagues [40] report that a one ounce equivalent increase in fresh lean beef by older Americans 65+ years was associated with a reduction in the odds of lower extremity mobility limitation by 22% (95% CI: 7%–34%) and any functional limitation by 15% (95% CI: 1%–28%). Similarly, research has suggested that protein and amino acids are positively associated with improved quality of life and well-being [41], and meat (defined by the authors as beef and veal, buffalo meat, pig meat, mutton and lamb, goat meat, horse meat, chicken meat, goose meat, duck meat, turkey meat, rabbit meat, game meat and offal) intake is positively correlated with increased life expectancy [42].

Although approximately 25% of beef consumers exceed the modeled level in the HDP, the excess is less than 2 ounces and does not result in protein consumption that exceeds the AMDR [30]. Based on the available modeled nutrient profiles, an additional 2 ounce equivalent of lean meat would provide an individual aged 19–70 years with an estimated additional 3 g of total fat, 1 g of saturated fat, and 40 mg of cholesterol, as well as 14 g of protein, 1 mg of iron, 2.3 mg of zinc, 52 mg of choline, 0.2 mg vitamin B6, and 1 mcg vitamin B12, if no other adjustments are made to the diet [15].

An et al. [43] previously reported no association between total beef consumption and overall dietary quality measured by the Healthy Eating Index (HEI) 2015 and no association between fresh lean beef consumption and daily intake of saturated fat. Further, fresh, lean beef consumption (2.0 ounce/day) was not associated with daily intake of total energy or sodium but was associated with increased choline, iron, and zinc intake when compared to non-beef consumers [43]. Nicklas et al. [44] reported that lean beef consumers consumed 4.4 ounces of lean beef daily yet consumed more servings of vegetables and fewer servings of milk, oils, grains and fruits than non-beef consumers with no significant difference in diet quality as measured by HEI. Compared to non-beef consumers, lean beef consumers were reported to consume the same amount of total fat with only 0.6% more saturated fat, 0.3% more monounsaturated fat, and 0.8% less polyunsaturated [44]. Therefore, based on these data, it is not expected that consumption of up to 2 ounces more than that modeled in the HDP would be of significant consequence for macronutrient intake and could benefit intake of micronutrients of dietary concern and/or challenge.

Of the individual beef types analyzed, processed beef was consumed the least. Recently, O’Connor and colleagues [45] reported similar findings in their assessment of intake patterns of fresh vs. processed red meat, noting that most of the total red meat consumed by Americans 2+ years was fresh (i.e., unprocessed). The majority of processed red meat consumed by Americans are pork-based with luncheon meat, sausage, ham, bacon, and hot dog representing the top five processed meats consumed [10]. The current data indicate that, on average, the majority of Americans 2+ years consume less than 0.25 ounces of processed beef per day.

Approximately 27% of individuals consumed ground beef on the day of recall. Ground beef is commonly consumed as burgers, as well as an ingredient in mixed dishes such as meatloaf, tacos, burritos, etc. which provide other important nutrients and is an affordable, versatile option for protein, but depending on the recipe, can also lead to increased sodium consumption due to other ingredients [7]. Our analysis concurs with a previous analysis of ground beef consumption patterns in Americans (using data from FoodNet Population Survey, 2006–2007) that noted that ground beef consumption decreases between adults (18–64 years of age) and older adults (≥ 65 years of age) [46]. Additionally, Taylor and colleagues similarly report that less than 40% of ground beef was consumed from fast food establishments [46].

A strength of the current study is the use of a large nationally representative population-based sample of children and adults to assess per capita and consumer-specific usual intakes of beef, using nine cycles (18-years) of NHANES data. Additionally, total beef as well as four individual beef types were separately analyzed to address common types of beef included in habitual diets to provide a fuller picture of beef consumption behaviors of Americans.

While dietary intake data such as NHANES provide direct estimates of intake, there are limitations. For example, NHANES results can be hampered by self-reporting bias [47], though it is expected that a large sample such as NHANES should provide a valid estimate of meat consumption with even a single 24-h recall [10]. On the other hand, national survey data and methods used to estimate meat consumption in the U.S. are varied and designed to meet particular data needs that may or may not meaningfully reflect actual beef intake [10,47,48,49].

Future NHANES-related research may consider the analysis of the contribution of beef and beef types to nutrient adequacy and/or diet quality of beef consumers and focus on specific vulnerable subpopulations and contribution to nutrient security. This study highlights the importance of the need for more accurate estimations of beef (and beef types) intake, which can increase rigor of future research to quantify the relationship between consumption of beef within broader dietary patterns and various health-related outcomes.

## 5. Conclusions

Beef is a nutrient-dense commonly consumed food that can contribute a variety of key essential nutrients that Americans may not otherwise consume in sufficient amounts from other foods [1,37]. Nearly every American has room to make improvements in their diet, and evidence-based dietary guidance is needed to support areas in need of improvements and that can benefit people the most. The current study analyzed consumption levels of total beef and beef types (fresh lean beef, ground beef, and processed beef) in Americans 2 years and older over an 18-year period (2001–2018) and found significant declines in beef consumption in children, adolescents, and adults, while consumption remained consistent in older adults. The average amount of total beef consumed increased from children and adolescents to adults but declined in older adults, which can impact aging-related health outcomes. Based on current beef intake data, beef consumers chose fresh lean beef in amounts typically within the MPE guideline, as modeled in the HDP at the 2000-calorie level. Given current beef intake trends, dietary guidance to limit or reduce beef intake could be viewed as lacking nutritional justification and could exacerbate the growing nutrient deficiencies in America. Beef is an inherent whole food source of high-quality dietary protein and several micronutrients including iron, zinc, and B vitamins; thus, a decline in beef intake can lead to unintended consequences of declining contribution of nutrients to the diet, including those identified as shortfall or essential nutrients [22,37,49]. Population-wide recommendations to increase or decrease the intake of any food group require evidence-based justification, given that all food groups provide critical nutrients that may not be readily available in meaningful amounts from other commonly available, popular, and/or affordable alternatives.

## Figures and Tables

**Figure 1 nutrients-15-02475-f001:**
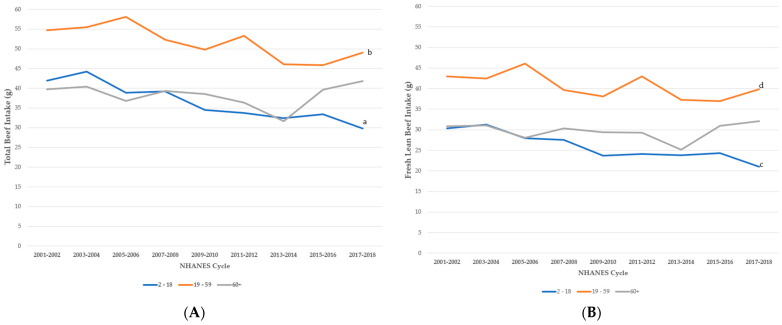
Changes over time in intake of (**A**) total beef; (**B**) fresh lean beef; (**C**) processed beef; and (**D**) ground beef in the United States, by age group. Regression analyses for beef intake over time with age, gender and race/ethnicity as covariates and NHANES survey cycle (2001–2018) as trend variable. Regression coefficient examines amount of change in g/day per cycle; *p*-value assesses whether regression coefficient is different from zero. ^a–g^ notes significant changes in intake between 2001–2022 and 2017–2018 cycle: ^a^ β = −1.66 *p* < 0.0001; ^b^ β = −1.14 *p* = 0.0004; ^c^ β = −1.22 *p* < 0.0001; ^d^ β = −0.68 *p* = 0.0206; ^e^ β = −0.33 *p* = 0.0005; ^f^ β = −0.39 *p* < 0.0001; ^g^ β = −0.80 *p* < 0.0001.

**Figure 2 nutrients-15-02475-f002:**
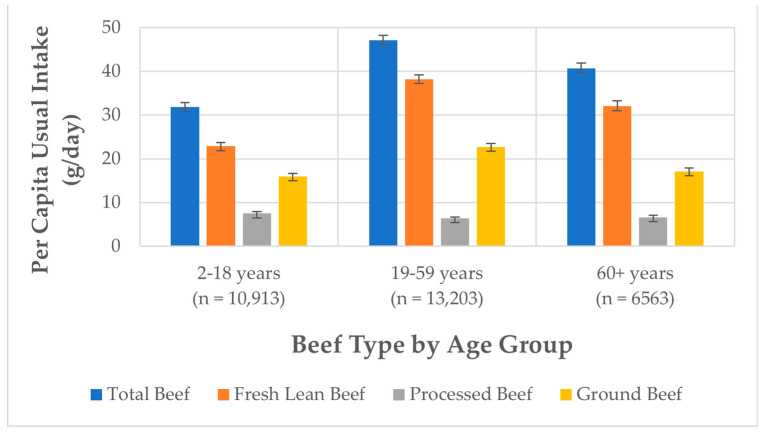
Per capita usual intake of total beef and beef types based on NHANES 2011–2018. Note: Beef types are not mutually exclusive, so the types will not equal the total, (e.g., fresh lean beef includes a portion that is ground, but not all ground beef is lean, and some consumers only consume certain beef types).

**Figure 3 nutrients-15-02475-f003:**
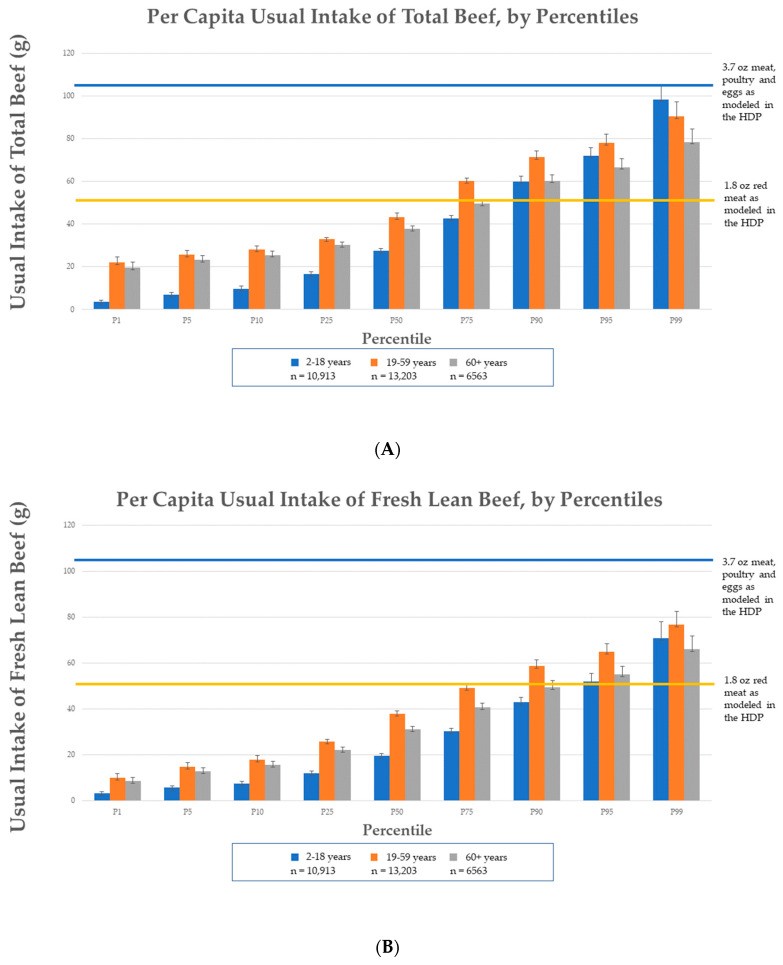
Percentiles of per capita usual intake of (**A**) total beef and (**B**) fresh lean beef based on NHANES 2011–2018 data by age group and compared to food groups modeled in the Healthy U.S.-Style Dietary Pattern (HDP) at the 2000-calorie level including red meat (i.e., fresh or processed beef, goat, lamb, pork, and game meat).

**Figure 4 nutrients-15-02475-f004:**
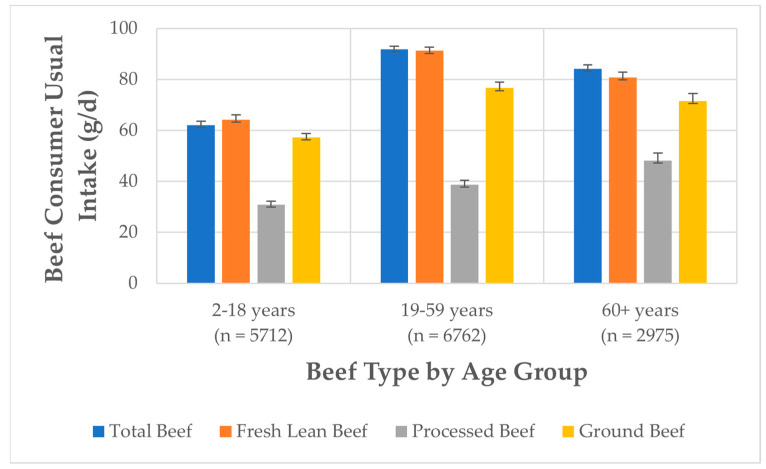
Beef consumer usual intake of beef types, by age group, based on NHANES 2011–2018. Note: Beef types are not mutually exclusive, so the types will not equal the total, (e.g., fresh lean beef includes a portion that is ground, but not all ground beef is lean, and some consumers only consume certain beef types).

**Figure 5 nutrients-15-02475-f005:**
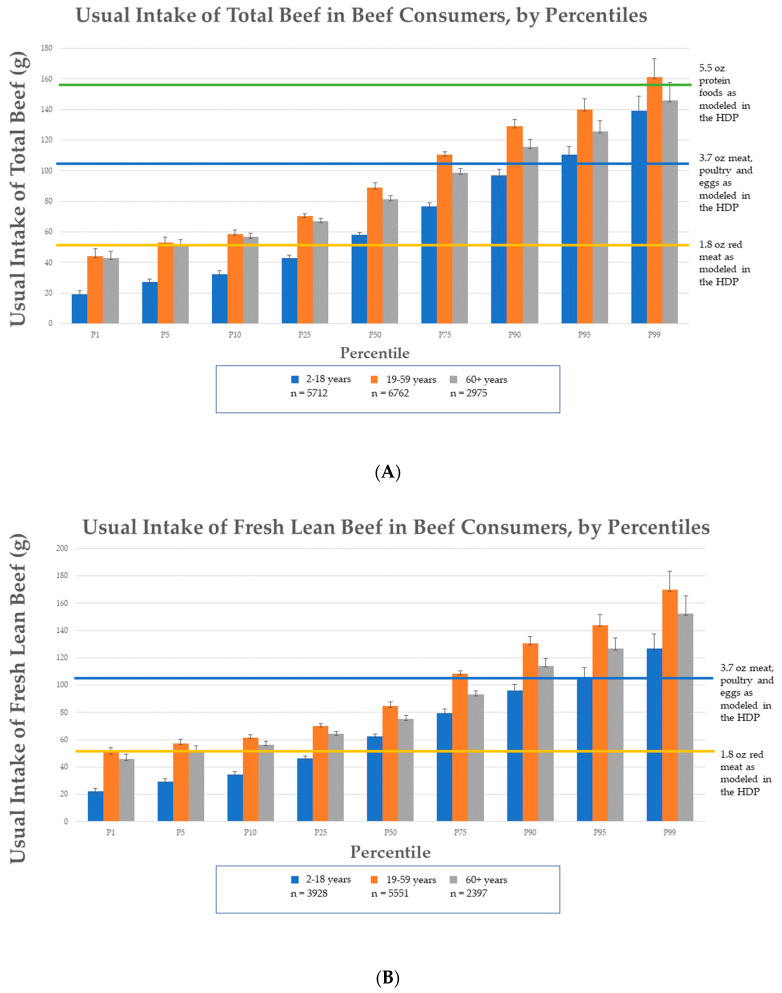
Percentiles of beef consumer usual intake of (**A**) total beef and (**B**) fresh lean beef based on NHANES 2011–2018 data by age group and compared to food groups modeled in the Healthy U.S.-Style Dietary Pattern (HDP) at the 2000-calorie level including red meat (i.e., fresh or processed beef, goat, lamb, pork, and game meat) and protein foods (i.e., meat, poultry, eggs, seafood, nuts, seeds, and soy products).

**Table 1 nutrients-15-02475-t001:** Beef-related Food Patterns Ingredient Database (FPID) Variables.

Beef Variable	Related FPID Variables	Calculation
Total Beef	pf_meat ^1^; pf_cured meat ^2^	Descriptions examined to determine the percentage of each variable assumed to be beef. Percentage of beef applied. Total includes solid fat in beef in excess of 2.63 g per ounce (9.28%) [i.e., discretionary fat] and lean beef portion.
Processed Beef	pf_cured meat	Cured meat includes frankfurters, sausages, corned beef, and luncheon meat. Descriptions examined to determine the percentage of each variable assumed to be beef. Percentage of beef applied. Total includes solid fat in beef in excess of 2.63 g per ounce (9.28%) [i.e., discretionary fat] and lean beef portion.
Fresh lean beef	pf_meat	Solid fat in excess of 2.63 g per ounce (9.28%) is subtracted, fresh lean portion remains
Ground beef	pf_meat; the descriptions of ingredients in FPID helped delineate ground beef products (i.e., ground beef or similar term used in food ingredient description)	Total includes solid fat in beef in excess of 2.63 g per ounce (9.28%) [i.e., discretionary fat] and lean beef portion.

^1^ pf_meat—FPID variable name “protein food–meat” includes beef, pork, veal, lamb, and game meat, excludes cured and organ meat. ^2^ pf_curedmeat—FPID variable name “protein-food-cured meat” includes frankfurters, sausages, corned beef, and luncheon meat made from beef, pork, or poultry.

## Data Availability

NHANES data are publicly available from https://www.cdc.gov/nchs/nhanes/index.htm accessed on 22 June 2021.

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
