# Peer review of "Trends in Beef Intake in the United States: Analysis of the National Health and Nutrition Examination Survey, 2001–2018"

_nutrients, 2023, doi:10.3390/nu15112475_

Round 1
Reviewer 1 Report
I commend the authors on a rigorous analysis of NHANES data that evaluated changes in beef intake over a nearly twenty-year period. NHANES data are commonly used to identify dietary trends and the methodology that was employed was sound. The distinction between total, lean, processed, and ground beef is helpful in light of data in recent years that has demonstrated differences in human health between processed and non-processed red meat. The results provide evidence against a common misconception that Americans consume more red meat than is recommended. The authors also point out the importance of dietary protein in the human diet, particularly among older adults. I have only a few suggestions to an otherwise strong analysis and well-written paper.
- While the focus of the paper was on intake patterns of beef, I think a few more citations surrounding recent large meta-analyses would be helpful to provide further context regarding the associations (or lack thereof, in some instances) between red meat intake and mortality (Zeraatkar et al Annals of Internal Medicine, 2019), overall health effects (Lescinsky et al Nature Medicine 2022), and mental health (Dobersek et al, Critical Reviews Food Science and Nutrition, 2021).
- While NHANES does not make the distinction between grass-fed and grain-fed beef, in addition to total, lean, processed, and ground beef in this study, I suggest a future direction of analyses evaluating changes in intake patterns between grass-fed and grain-fed beef in light of some of the differences between the two in fatty acid content, (Nogoy et al Food Sci Animal Resource, 2022).
I commend the authors on this work that provides up to date evidence using reputable NHANES data on beef intake patterns in the United States, a topic of considerable debate and common misperceptions.
Author Response
Reviewer Comment: While the focus of the paper was on intake patterns of beef, I think a few more citations surrounding recent large meta-analyses would be helpful to provide further context regarding the associations (or lack thereof, in some instances) between red meat intake and mortality (Zeraatkar et al Annals of Internal Medicine, 2019), overall health effects (Lescinsky et al Nature Medicine 2022), and mental health (Dobersek et al, Critical Reviews Food Science and Nutrition, 2021).
Response: Thank you for the suggestions. A paragraph has been added starting line 354 specifically discussing the citations above, with the exception of Dobersek et al, as the relationship between meat consumption and mental health outcomes is still emerging and not often discussed in the context of dietary guidance regarding meat intake. Please see attachment for revised manuscript.
Reviewer Comment: While NHANES does not make the distinction between grass-fed and grain-fed beef, in addition to total, lean, processed, and ground beef in this study, I suggest a future direction of analyses evaluating changes in intake patterns between grass-fed and grain-fed beef in light of some of the differences between the two in fatty acid content, (Nogoyet al Food Sci Animal Resource, 2022).
Response: Thank you for the suggestion, however we believe since this is outside the scope of the available data in NHANES, it is outside scope of future directions we’re comfortable suggesting in the current manuscript.

Reviewer 2 Report
Title: Trends in Beef Intake in the United States: Analysis of the National Health and Nutrition Examination Survey, 2001–2018
The manuscript “Trends in Beef Intake in the United States: Analysis of the National Health and Nutrition Examination Survey, 2001–2018” characterized the intake trends for total beef and specific beef types among Americans participating in the National Health and Nutrition Examination Survey 2001-2018 nd usual intake was assessed. The study concluded that beef is not overconsumed by the majority of Americans. It is well written article with some interesting findings; however, there are some corrections before its acceptance for publication:
Line 21: Please specify the age group here “Americans 2+ years…”.
Line 26: I would suggest to add concluding remarks at the end of the abstract. The last line (line 25-26) may be extended: “Evidence from intake trends suggests beef is not overconsumed by the majority of Americans…”.
The introduction part in well written, however, I would suggest to add a paragraph on required vs current (different types) meat intake and may be explained it with disease occurrence i.e., health risks associated with meat consumption in US. I would suggest to read and cite the following references in that paragraph:
· doi: 10.1017/S1368980010002077
· DOI: 10.1024/0300-9831/a000224
· doi: https://doi.org/10.1136/bmj.m4141
· DOI: https://doi.org/10.1186/s12916-021-01922-9
Line 76: The link (https://www.cdc.gov/nchs/nhanes/about_nhanes.htm) is not working, please check and revise.
Line 82: Why authors choose the 19-59 years age group, as it may involve some other changes involved with physiological state of the body? How authors would explain or justify this age group.
Line 174: What is NCI pro-grams?
Results parts is well explained; however, discussion part needs some improvement and I would suggest the authors to use some references and relate the meat intake with any diseases such as cardiovascular diseases and obesity conditions among the population.
Line 470: Keeping in view the importance of the topic, authors should suggest some guidelines for future research, i.e., which aspect should be focused for future research.
Author Response
Please see attachment for revised manuscript.
Reviewer Comment: Line 21: Please specify the age group here “Americans 2+ years…”.
Response: We have added clarification to the sentence on line 21 in revised manuscript.
Reviewer Comment: Line 26: I would suggest to add concluding remarks at the end of the abstract. The last line (line 25-26) may be extended: “Evidence from intake trends suggests beef is not overconsumed by the majority of Americans…”.
Response: We have added more context to concluding remarks in the abstract.
Reviewer Comment: The introduction part in well written, however, I would suggest to add a paragraph on required vs current (different types) meat intake and may be explained it with disease occurrence i.e., health risks associated with meat consumption in US. I would suggest to read and cite the following references in that paragraph:
- doi: 10.1017/S1368980010002077 [Daniel CR, Cross AJ, Koebnick C, Sinha R. Trends in meat consumption in the USA. Public Health Nutr. 2011 Apr;14(4):575-83.]
- DOI: 10.1024/0300-9831/a000224 [Battaglia Richi E, Baumer B, Conrad B, Darioli R, Schmid A, Keller U. Health Risks Associated with Meat Consumption: A Review of Epidemiological Studies. Int J Vitam Nutr Res. 2015;85(1-2):70-8.]
- doi: https://doi.org/10.1136/bmj.m4141 [Al-Shaar L, Satija A, Wang DD, Rimm EB, Smith-Warner SA, Stampfer MJ, Hu FB, Willett WC. Red meat intake and risk of coronary heart disease among US men: prospective cohort study. BMJ. 2020 Dec 2;371:m4141.]
- DOI: https://doi.org/10.1186/s12916-021-01922-9 [Papier K, Fensom GK, Knuppel A, Appleby PN, Tong TYN, Schmidt JA, Travis RC, Key TJ, Perez-Cornago A. Meat consumption and risk of 25 common conditions: outcome-wide analyses in 475,000 men and women in the UK Biobank study. BMC Med. 2021 Mar 2;19(1):53.]
Response: Thank you for the suggestion. Please note that in the introduction section, we have included a more recent assessment than the suggested Daniel et al 2011 publication of meat consumption trends - see reference 10 (Zeng et al 2019) on line 57 in revised manuscript. Rather than adding to introduction, we have added the two most recent suggested citations to a new paragraph in the discussion section, as recommended by reviewer 1, see line 354 in revised manuscript.
Reviewer Comment: Line 76: The link (https://www.cdc.gov/nchs/nhanes/about_nhanes.htm) is not working, please check and revise.
Response: Link works as of May 17, 2023.
Reviewer Comment: Line 82: Why authors choose the 19-59 years age group, as it may involve some other changes involved with physiological state of the body? How authors would explain or justify this age group.
Response: The age groups analyzed in the current manuscript are based on the life stages as outlined in the Dietary Guidelines for Americans, 2020-2025: children and adolescents ages 2 through 18 years, adults ages 19 through 59 years, and older adults age 60 years and older. Clarification has been added on Line 85 in revised manuscript.
Reviewer Comment: Line 174: What is NCI programs?
Response: Thank you for catching this. NCI is abbreviation for the National Cancer Institute. We have added abbreviation on line 90 in revised manuscript.
Reviewer Comment: Results parts is well explained; however, discussion part needs some improvement and I would suggest the authors to use some references and relate the meat intake with any diseases such as cardiovascular diseases and obesity conditions among the population.
Response: Thank you. This feedback was aligned with reviewer 1. A new paragraph in the discussion section has been added, starting at line 354 in revised manuscript
Reviewer Comment: Line 470: Keeping in view the importance of the topic, authors should suggest some guidelines for future research, i.e., which aspect should be focused for future research.
Response: Thank you. A paragraph on future recommendations has been added starting line 459 in revised manuscript

Round 2
Reviewer 2 Report
The manuscript is sufficiently improved according to the comments and suggestions of the reviewer and may be accepted in present form.